# Review on Abrasive Machining Technology of SiC Ceramic Composites

**DOI:** 10.3390/mi15010106

**Published:** 2024-01-07

**Authors:** Huiyun Zhang, Zhigang Zhao, Jiaojiao Li, Linzheng Ye, Yao Liu

**Affiliations:** School of Mechanical Engineering, North University of China, Taiyuan 030051, China; szys2057324@126.com (H.Z.); zhao105820@163.com (Z.Z.); lijiaojiao1004@gmail.com (J.L.); yelinzheng@nuc.edu.cn (L.Y.)

**Keywords:** ceramic matrix composites, material removal mechanism, numerical simulation, abrasive machining

## Abstract

Ceramic matrix composites have the advantages of low density, high specific strength, high specific die, high-temperature resistance, wear resistance, chemical corrosion resistance, etc., which are widely used in aerospace, energy, transportation, and other fields. CMCs have become an important choice for engine components and other high-temperature component manufacturing. However, ceramic matrix composite is a kind of multi-phase structure, anisotropy, high hardness material, due to the brittleness of the ceramic matrix, the weak bonding force between fiber and matrix, and the anisotropy of composite material. Burr, delamination, tearing, chips, and other surface damage tend to generate in the machining, resulting in surface quality and strength decline. This paper reviewed the latest abrasive machining technology for SiC ceramic composites. The characteristics and research directions of the main abrasive machining technology, including grinding, laser-assisted grinding, ultrasonic-assisted grinding, and abrasive waterjet machining, are introduced first. Then, the commonly used numerical simulation research for modeling and simulating the machining of ceramic matrix composites is briefly summarized. Finally, the processing difficulties and research hotspots of ceramic matrix composites are summarized.

## 1. Introduction

Ceramic matrix composites (CMCs), mainly including SiC_f_/SiC, SiC_p_/SiC, C_p_/SiC, and C_f_/SiC, have the advantages of low density, high specific strength, high elastic modulus, high temperature resistance, wear resistance, and chemical resistance [1]. As the result of fiber reinforcement, the CMC materials have improved fracture toughness and are widely used in aerospace, energy, transportation, and other fields to resist the thermal shock in high temperature environments. Additionally, CMCs are lightweight (about 1/3 of traditional metal materials) and have high combustion temperature, thus offering significant advantages in terms of lightweight design, fuel efficiency, and the gas emissions of aviation engines, and making them the preferred materials for high-temperature components of aviation engines. 

As illustrated in Figure 1a, the use of CMC in blades, guides, hoods, nozzles, and combustion chambers in high-temperature sections of an aeroengine, increases engine thrust by 25%, saves fuel consumption by 10%, and decreases nitrogen oxide emissions. As shown in Figure 1b, the CMCs are used to manufacture non-structural parts such as thermal barrier coating and thermal conductive sheets of the engine, and key structural components, such as engine blades, guides, nozzles, and combustion chambers, to reduce the weight of the whole engine and to improve its durability and life. In Figure 1c, in Japan’s 100 kW automotive ceramic gas turbine (CGT) project, CMCs were used to manufacture five components, including turbine rotor, rear plate, port bushing, expansion refiner, and internal scroll support [2]. It has demonstrated the feasibility of the application of CMCs in operational engines and their superiority in terms of thermal shock resistance and particle impact resistance capabilities. 

CMCs are very difficult to machine due to the complex woven structure, anisotropic properties, and high hardness. CMC components are usually fabricated in a rough mold and are then machined to the design dimension, shape accuracy, and surface condition. Burr, delamination, crack, collapse, tear, and other surface damage easily occur, resulting in a machining quality decline, which creates a challenge for CMC machining. Simultaneously, the material removal mechanism of CMCs is very complicated, which seriously limits research progress. Fiber in CMC not only enhances the mechanical properties of the matrix, but also changes the material removal mechanism. In the grinding process, the damage usually starts from the matrix. The different material properties of the matrix and fiber show different removal modes, and the interactions transferred through the interface affect each other. Therefore, exploring the material removal mechanism is an effective way to realize lower damage and greater efficient machining of the CMC materials. 

In this paper, the recent abrasive machining experimental study of CMC abrasive machining technologies, including the conventional grinding (CG), laser-assisted grinding (LAG), abrasive waterjet machining (AWJM), and ultrasonic vibration-assisted grinding (UVAG), were reviewed to reveal the material removal mechanism. Then, the CMC simulation studies, including FEM, SPH, DEM, and a hybrid of two or three methods, were introduced. Finally, the challenging aspects and pivotal points of the abrasive machining mechanism are summarized.

## 2. Experimental Study of CMC Abrasive Machining

To meet the requirements of complex shapes and dimensional tolerances, CMC materials need to be machined. To improve the machining surface integrity and manufacturing efficiency of CMC, research was conducted, both domestically and abroad. Abrasive machining is the most suitable method for the precision machining of CMC materials [3]. This section introduces the latest achievement in conventional grinding (CG), laser-assisted grinding (LAG), abrasive waterjet machining (AWJ), and the ultrasonic vibration-assisted grinding (UVAG) of CMCs.

### 2.1. Conventional Grinding

CG is the most common method for the precision machining of CMC [4]. The ground surface characteristics and material removal mechanism of CMCs have been analyzed. In the CMC grinding process, the orientation of fibers exhibits a crucial influence on the ground surface quality [5]. Qu et al. [6] studied the effects of carbon fiber orientation and grinding parameters on grinding force and surface quality during the grinding of C_f_/SiC, as shown in Figure 2a. The results indicated that grinding depth has significant effects on the surface quality. Specifically, the surface quality decreases and the grinding forces increase with increasing grinding depth. In addition, greater grinding surface quality is observed at *β* = 90°. The poorer ground surfaces are obtained at *α* = 0°. Cao et al. [7] studied the grinding process of woven CMC. The results showed that the best surface quality is in the 90° direction with the fiber. Hu et al. [8] concluded that the local orientation of fibers would also affect the surface quality of processed materials. Yin et al. [9] used the single grain scribing on SiC_f_/SiC with different wheel speeds to reveal the material removal mechanism and fiber breakage behaviors. The results showed that the surface material densification and smearing could be suppressed by increasing grinding speed. When grinding along the fiber longitudinal direction, fibers experience plowing and would be covered by the smearing of the matrix. Increasing wheel speed enhances the brittle fracture and breakage of the fibers. In high-speed grinding, fibers present brittle fractures and the matrix is torn off. When grinding transverses to the fiber longitudinal direction, increasing wheel speed leads to the complete removal of the fibers and a few cutting-off fiber end residuals on the groove bottom surface, which improves the surface finish. Zhang et al. [5] compared the surface morphology and grinding mechanism of C_f_/SiC composites in transverse, normal, and longitudinal directions (in Figure 2b), and found that the fibers were most easily removed during transverse grinding, and the ground surface quality was the best in this direction. During normal direction grinding, fibers and matrix are mainly removed in the form of small fragments, and the ground surface quality is worse than that in the transverse direction. In longitudinal grinding, the composite material has the worst ground surface quality. Liu et al. [10] proposed a three-dimensional surface profile characterization through experiments, which can describe the surface quality of composite materials accurately.

Liu et al. [11] investigated the effect of fiber angle (FA) on grinding force, surface morphology, and roughness through surface grinding experiments. As shown in Figure 2c, the fracture characteristics of the grinding surface include matrix cracking, fiber fracture, and interfacial debonding, revealing that the main removal mechanism of 2D C_f_/SiC. Fiber fracture was more severe at 30° and 45° FA than at 0° FA. Those differences can be attributed to the fact that the damage is not synchronized during the grinding process due to the different mechanical properties of the matrix and fibers. Figure 2d shows the surface morphology of 0° and 90° fiber bundles at different grinding angles. During the grinding process, the scraping force shows a periodic variation of 0°and 90° fiber weave structure when the scraping angle is 0°. However, when the scribing angle is 45°, the force is relatively more stable and the surface quality is better.

In the grinding of CMCs, grinding depth, feed speed, friction force, lubricant, and tool parameters have important effects on surface quality and fracture characteristics. Wang et al. [12] established a force model for the surface grinding of unidirectional C_f_/SiC composites, and concluded that the grinding force is inversely proportional to the wheel speed and proportional to the grinding depth and feed speed. Zhang et al. [13] established a model between grinding force and depth in fiber-reinforced composites. With the increase of grinding depth, the grinding force exhibits an almost linear relationship along the direction of the reinforced fiber, which affects the ground surface quality. Zhang et al. [5] performed the experimental research on the grinding mechanism of woven CMC and revealed that both the grinding force and the surface roughness increased with the feed speed and the grinding depth, and decreased with the wheel speed. Tawakoli et al. [14] investigated the CMC grinding forces and surface quality during conventional and intermittent grinding by using the normal and fan-shaped wheel. The results found that the conventional grinding wheel could obtain better surface roughness and the fan-shaped grinding wheel could significantly reduce the grinding force and grinding temperature. These results revealed that intermittent grinding could reduce scratching, the plowing phenomenon, and specific energy, which can provide a high surface quality. 

The CMC grinding damage modes can be analyzed through indentation fracture mechanics [15]. Yang et al. [16] carried out orthogonal grinding of C_f_/SiC to observe the ground surface topography and found that the main material removal mode was brittle fracture, including matrix cracking, fiber break, fiber wear, and interface bonding. Pineau et al. [17] built a machining force prediction model based on the experimental results of the shear strength and predicted the crack occurrence of woven CMC based on the virtual test results. Lamon et al. [18] proposed the main damage mode of two-dimensional woven C_f_/SiC based on the micromechanical method of the mechanical behavior of brittle matrix composites.

Due to the high hardness and low thermal conductivity of C_f_/SiC composites, the grinding wheel absorbs and stores a lot of heat during dry grinding, resulting in the reduction of grain number, sharpness, and life. In addition, due to the passivation of the grinding wheel, the surface quality of the machined surface decreases [19]. Hu et al. [20] studied the cutting force, surface integrity, and machining defects in the milling process of 2D C_f_/SiC composites with the commercialized PCD tool. The results showed that fiber fracture, matrix damage, and fiber-matrix debonding are the main types of material failure modes. Xu et al. [21] studied the dry cutting temperature of C_f_/SiC with a straight-edge PCD tool. The study showed that the highest temperature reached around 732 °C, with a significant increase in cutting force, which makes the surface quality poor. Qu et al. [22] investigated the effect of minimum quantity lubrication (MQL) on the grinding performance of unidirectional carbon fiber-reinforced ceramic matrix composites, as shown in Figure 3a, which includes abrasive, lubricant, and material. The surface morphology was observed using scanning electron microscopy (in Figure 3b). The main fracture patterns of carbon fiber are smooth separation, fracture, detachment, and pulling out. 

CMC grinding has shown serious tool wear, low machining efficiency, and poor machining quality. In particular, defects such as poor dimensional consistency, surface cracks, fragmentation, delamination, fiber shedding, fiber pulling out, and chipping seriously affect the processing quality and surface accuracy of the material [23]. Therefore, in the grinding process, the appropriate grinding depth and cutting angle should be carefully selected.

### 2.2. Laser-Assisted Grinding (LAG)

Due to the high brittleness and anisotropy of the CMC, LAG acts as a feasible process with which to improve the machinability of materials, which induces the strength loss of materials through high temperatures. LAG uses the thermal effect of the laser to soften the material in the area to be ground, which can reduce the cutting force and improve the machinability of the material. Compared to traditional processing, LAG can reduce the force, surface roughness, and production costs.

Zhang et al. [24] designed a single-factor experiment to understand the micro-structure evolution and ablation behaviors of the microgrooves processed by a picosecond laser. This study provided a theoretical basis and practice guidance for LAG of the SiC_f_/SiC composites. Figure 4a shows the amorphous SiO_2_ smoke, with extremely fine particles. The dust attached to the treated surface caused the occurrence of pileup, leading to a slight increase in Ra value along the machined microgrooves. When the laser parameters (pulse energy, repetition frequency, scanning times, and scanning speed) were adjusted to increase the microgroove depth, the SiC vapored in the subsurface layer was recrystallized during rapid cooling, as presented in Figure 4b. Compared to the ultrahigh hardness SiC fiber and SiC matrix, the SiO_2_ smoke dust or recrystallized SiC were powdery due to the ablation, which were easy to remove and were expected to greatly improve the surface quality. An et al. [25] analyzed the products of continuous laser ablating CMC, and the factors affecting the depth of ablation layer were analyzed. As shown in Figure 4c, according to the morphology and types of ablation products, the laser ablation of SiC_f_/SiC can be divided into the coagulated layer (Layer 1), the re-crystalized SiC layer (Layer 2), the heat-affected layer (Layer 3), and the non-affected layer (Layer 4). In Layer 1, gaseous SiC and oxygen in the air undergo an active oxidation reaction at high temperatures to form smoky amorphous SiO_2_ condensation deposited on the surface of the ablated layer. Layer 2 is mainly composed of re-crystalized SiC particles with a micro-size scale. A small amount of SiO_2_ soot is attached to the particles.

As shown in Figure 5, Zhou et al. [26] proposed a laser-induced ablation-assisted grinding (LIAAG), which is based on the chemical properties of materials. This method utilizes lasers to ablate workpieces before grinding, aiming for high efficiency, minimal damage, and reduced abrasive wear, and found that C_f_/SiC composites were chemically transformed into relatively loose and homogeneous ablation products (SiO_2_ and recrystallized SiC) at high laser ablation temperatures. In Figure 5c, surface morphologies displayed the microfracture and crushing of carbon fibers and SiC matrixes, and the grinding-induced damages, such as macro fracture, fiber pulling out, and interface debonding. In Figure 5f, the abrasive belt was primarily worn in micro-adhesion and micro-abrasion, rather than cleavage fracture and fall-off in traditional grinding. The surface integrity was improved greatly, and the abrasive wear was reduced significantly, which provided a vital high-performance processing method for CMC components. 

Li et al. [27] introduced the laser-assisted precision grinding technology to improve the processing quality of 3D woven C_f_/SiC composites. The laser process parameters were adjusted to control the depth of the thermally induced damage layer and to reduce the hard brittleness of the material. In the study, experiments were carried out to investigate the effect of laser parameters on material damage and the effect of LAG processes. As shown in Figure 6a, due to the material anisotropy, the depth and width of the thermal-damage slot are 0° fiber > 90° fiber > normal fiber and 90° fiber > 0° fiber > normal fiber, respectively. In addition, laser irradiation causes complex reactions, such as sublimation, decomposition, and oxidation, on the surface of the material and generates SiO_2_ and Si as well as recrystallized SiC, resulting in the formation of a porous SiC layer (thermal metamorphic layer) on the subsurface, as shown in Figure 6b. In the LAG, the normal grinding force, the tangential grinding force, the specific grinding energy, and surface roughness are reduced by a maximum of 35.6%, 43.6%, 43.58%, and 24.22%, respectively, compared to the conventional grinding (CG) process. As shown in Figure 6c–e, the material removal of 3D C_f_/SiC composite material is dominated by brittle removal, which is mainly dominated by fiber breakage, pulling out, layer breakage, matrix cracking, and interface debonding. However, a trend toward ductile domain removal has begun to emerge.

Kong et al. [28] investigated the cutting performance of CG and LAG of SiC_f_/SiC using electroplated diamond grinding heads. A three-dimensional transient heat transfer model based on a Gaussian heat source was developed to investigate the distribution of temperature fields on both the surface and subsurface of SiC_f_/SiC subjected to laser irradiation. As seen in Figure 7, after laser irradiation, the surface and subsurface temperatures of the workpiece reach > 1000 °C, which is sufficient for the oxidation reaction and softening of the material. In Figure 7c,d, the effects of laser heating temperature on the workpiece surface on the grinding forces were analyzed. The axial and feed grinding forces were more than 40% lower under LAG than CG, due to the removal mechanism of the SiC matrix changing from brittle to ductile and the oxidation reactions occurring in the SiC_f_/SiC composites. The material removal mechanism was analyzed by observing the morphology of machined surfaces, as shown in Figure 7e–g, which showed that ductile removal from the SiC matrix occurs during LAG. In terms of abrasive wear, the mean height of exposed abrasive grains from the machined surface was reduced by 1.02 μm and 12.52 μm in LAG and CG, respectively. Under CG, the abrasive grains mainly exhibit cleavage fractures; however, under LAG, micro-abrasion is the main wear form.

LAG can reduce the CMC surface temperature gradient and hardness, which improves the grinding conditions and achieves a smooth ground surface. After laser preheating, the surface temperature of the processed material increase, and the hardness of the material is reduced. To avoid rapid temperature reduction, it is necessary to minimize the time difference between the laser and grinding.

It can be seen from the above research that the grinding force of the CMC is significantly reduced under laser heating. The tool wear and machining defects are also reduced. However, due to CMCs with high melting points and high hardness, the temperature required for material softening, melting, and even gasification is very high—higher than 1000 °C. Therefore, the laser heat-affected zone is large, in which the physical and mechanical properties of the CMC are changed. The material in the heat-affected zone must be removed to obtain the required surface. Moreover, interface cracks and surface oxidation caused by the laser thermal effect made LAG difficult to apply to the shape machining of typical components of complex curved surfaces [29]. 

### 2.3. Abrasive Waterjet Machining (AWJM)

AWJM is a new technology that has been developed rapidly in the past 30 years. It uses high-pressure and high-speed water jets to impact the workpiece to achieve cutting, perforating, and surface material removal [30]. For CMCs, AWJM technology has unique advantages. It belongs to the category of non-contact processing and avoids the problem of tool wear. In addition, the processing temperature is low, which can greatly reduce the thermal effects zones. The study by Hashish et al. [31] shows that the AWJM can successfully cut CMCs. However, there are still several problems that exist, such as serious nozzle wear, and micro and macro defects.

Hashish et al. [32] cut the SiC_f_/SiC with AWJM at different impact velocities, and found that the size of the edge breakage decreased with the impact velocity decrease. The taper decreases with the increase of abrasive hardness and injection pressure, and increases with the cutting speed increase. In addition, there is a parallel relationship between corrugation and taper. The influences of abrasive hardness, injection pressure, and cutting speed on corrugation are similar to those on taper. In addition, by comparing the influence of #80 and #120 abrasive grain on the edge breakage, it was found that the small grain size can reduce the edge breakage defect. 

Zhang et al. [33] explored the accurate control of hole shape for AWJM hole-making of C_f_/SiC based on experimental and mathematical analysis methods, as shown in Figure 8. The results reveal that D_difference_ is influenced by the standoff distance, followed by the traverse speed. However, influence of the pressure and the abrasive flow rate is rare. The traverse speed, pressure, and abrasive flow rate affect the D_difference_ by changing the total energy of the jet. The standoff distance mainly affects the D_difference_ by changing the effective impact area, which is fundamentally different from other process parameters. In Figure 8b, when the jet is cutting circular trajectories, there is a deflection effect, which causes D_jet_ to be larger than D_trajectory_ and affects hole size accuracy. The deflection effect increases with the increase of traverse speed and standoff distance and decreases with the increase of waterjet pressure and D_trajectory_. Therefore, when holes are cut at high traverse speed, low pressure, and a tiny trajectory diameter, the deflection effect must be considered. However, this study does not relate to surface damage in composite materials processing.

Ramulu et al. [34] researched the cutting force and surface micro-structure of composite materials, and material tensile behavior, under AWJM machining and conventional milling processing. The results showed that AWJM machining exhibits superior stability in the machining process and yields better surface quality. The cutting force is much lower than that of CG. Ren et al. [35] summarized the advantages and mechanism of AWJM machining ceramics and found that surface roughness could be controlled by controlling the relationship between the fibers’ axial direction and the AWJM direction. When the AWJM direction is parallel to the axis of the fibers, at the top of the drilling hole, the material processing surface is relatively flat, and the fracture surface of the fiber and the matrix is consistent under the action of the AWJM. Inside the borehole, prominent broken fibers were seen, and the SiC matrix between the fibers was removed and carried away by the AWJM. At the exit of the borehole, the fibers pull out, break, and strip from the matrix. When the direction is perpendicular to the axial direction of the fibers, the drilling hole is relatively flat from the top to bottom. The matrix around the fiber is not spalling, and the fiber pulls out. However, the fiber and the matrix threshing phenomenon is reduced.

Generally, AWJM has many advantages, such as the material not being affected by heat during processing, which can effectively improve the quality of in-hole processing. However, when processing deeper holes, the surface of the workpiece is prone to burr and chipping. The entrance and exit dimensional error may be large and the processing quality is relatively low [36]. 

### 2.4. Ultrasonic Vibration-Assisted Grinding (UVAG)

Ultrasonic processing technology employs an ultrasonic generator to convert electrical energy into ultrasonic waves that oscillate at a specific frequency and vibrate through an amplification tool (variable amplitude bar). The ultrasonic wave was generated in UVAG. The suspended particles in the working fluid impact the surface of the workpiece and remove excess material. UVAG can machine the insulation material and complicated three-dimensional structures, regardless of material hardness. To solve the problems of poor machining quality and serious tool wear in CMC material processing, researchers both domestically and aboard have conducted comparative experiments between UVAG and CG to observe and test its technological parameters.

Kang et al. [37] experimented with the UVAG of SiC_f_/SiC material on the end face of the diamond grinding wheel. The experimental setup and operational principles are shown in Figure 9, aimed at examining the removal mechanism, grinding force, surface morphology, and surface roughness. The results show that the appropriate ultrasonic amplitude can effectively reduce the grinding force, induce the fracture of SiC fibers, and improve the surface finish. However, when the amplitude is too large, the surface impact is large, and the surface quality declines. Ding et al. [38] studied the surface/subsurface breakage formation mechanism and machining quality of C/SiC composite conducted by UVAG and CG tests. The results showed that main breakage types of different angle fibers in ground surface were lamellar brittle fracture. Compared to CG, these breakages were reduced by UVAG which can reduce grinding force. Moreover, the ground surface roughness obtained by UVAG was lower than CG and the maximum reduction was 12%. Zhang et al. [39] investigated the material removal and breakage mechanism in UVAG of two-dimensional woven carbon-fiber-reinforced silicon carbide matrix composites (2D-C_f_/SiC). The results show that the predominant material removal mode in UVAG is brittle fracture. The forms of material breakage are matrix cracking, fiber fracture, fiber pull-out, interfacial debonding, and interfacial fracture. Compared with CG, the normal force, tangential force, and surface roughness in UVAG decreased by approximately 20%, 18%, and 9%, respectively. Bertsche et al. [40] studied the material removal rate, cutting force, tool wear, and surface roughness of rotary ultrasonic machining of CMC. Compared to conventional cutting processes, UVAG effectively reduced cutting force and tool wear, and significantly improved surface roughness. Huang et al. [41] compared six commonly used tools for micro-hole drilling of SiCf/SiC CMC assisted by ultrasonic vibration: carbide drill, PCD drill, electroplated diamond abrasive tool, grinding drill, coated grinding drill, and PDC tool. As shown in Figure 10, the influence of different kinds of tools on cutting forces, machining accuracy, and tool wear was investigated. The results showed that the machining accuracy of the PDC tool is best, followed by the electroplated diamond abrasive tool. The machining accuracy of the grinding drill and coated grinding drill is poor with a bit of difference; PDC and coated grinding drill have less tool wear, while the grinding drill has slightly more severe tool wear. However, the tool wear of the electroplated diamond abrasive tool is severe. Wang et al. [42] conducted rotary ultrasonic machining of C_f_/SiC composites to analyze the micro-structural characteristics of the hole surfaces under various fiber directions, ultrasonic amplitudes, and spindle speeds. The results show that the fiber direction and spindle speed have significant effects on the surface morphology of the material. The introduction of ultrasonic vibration improves the surface quality of C_f_/SiC composites, and higher ultrasonic amplitude and lower spindle speed are helpful for improving the surface quality of the hole.

Under the same conditions, UVAG can reduce the cutting force and cutting temperature, which can effectively reduce the chip, burr, crack, fiber stripping, and processing damage of CMCs, and improve the processing precision and quality. Therefore, UVAG technology has been widely used in the precision machining of CMCs. And it is mainly suitable for the processing of small holes or micro-structures. However, as the interaction between tool and material in UVAG becomes more complex, the realization of the judgment and control of material removal mode is difficult. Moreover, the aggravated wear of the tool induced by high-frequency micro-vibration and the very low efficiency of large-diameter ceramic holes processing are the main drawbacks of UVAG [23,43].

### 2.5. Summary for Experiment Study

In summary, as shown in Table 1, compared to CG, non-CG technologies have obvious advantages. However, there are also certain limitations. For example, LAG requires strict control of technical parameters such as optimal power; UVAG is limited by ultrasonic critical speed, and the processing efficiency is low. AWJM in the processing of deep holes appear as a large dimensional error and surface burr, and the chip phenomenon.

Compared with the single special processing technology, the composite processing technology can show more excellent results. At present, several composite machining technologies have been developed, such as WJM and laser composite machining technology [44], ultrasonic and EDM composite machining technology [45], laser and ellipsoid ultrasonic vibration composite machining technology [46], ultrasonic vibration and electrolytic linear dressing (ELID) grinding composite machining technology [47], and other new composite machining technologies. These technologies have their advantages. There is still a lack of a process method that can simultaneously take into account the surface quality, efficiency, and cost of CMC processing. Therefore, the theoretical research of the new multi-energy field composite machining method with high efficiency and quality and its applications are still the research hotspots in the field of aerospace high-tech manufacturing.

## 3. Simulation Study of CMC Abrasive Machining 

The microscopic results of CMC materials’ processing can be obtained directly through experimental testing, such as the machinability, tool wear, forces, and surface quality. And with the aid of microscopes, subsurface damages induced by machining can also be revealed. Therefore, extensive experimental works on CMC machining have been conducted in the past decades to investigate and understand the process of machining composites. However, experimental investigations cannot explore the instant occurrence of material removal, tool-matrix-fiber phase interaction, and damage during grinding. This has significantly limited the depth of understanding and the level of process design. In addition, micro-structural examinations and experimental investigations are expensive and time-consuming. The advances in numerical analysis enabled comprehensive studies of composite grinding through the mechanic’s model. The numerical techniques used to study the CMC grinding process mainly include the finite element method (FEM), smoothed particle hydrodynamics (SPH), molecular dynamics (MD) analysis, and discrete element method (DEM) analysis.

### 3.1. Finite Element Method (FEM)

The finite element method (FEM) studies the real physical systems’ (geometry and load states) interaction through numerical analysis. The dynamic explicit algorithm is used in FEM due to the extreme deformation and complicated interactions among the matrix, fiber, interface, and tools in CMC machining. Grinding is performed continuously by tiny, abrasive cutting edges. Experimental observation and analysis of the grinding process is very difficult due to the large number of grains, irregular geometry, high grinding speed, and small and inconsistent grinding depth. The use of FEM can save a lot of time when determining experimental processing parameters. The FEM can obtain the ground surface morphology, force, temperature distribution, chip formation, and coupling relationship of parameters.

Zhang et al. [48] studied the mechanical properties of C_f_/SiC composites and the influence of interface phase on mechanical properties by using a cohesion model and the Oliver–Pharr method. The interface strength and thickness influence on the load-displacement curve during nanoindentation was analyzed. It was found that, due to the presence of the interface, the in-situ mechanical properties of carbon fiber materials were changed. The hardness and elastic modulus of carbon fiber materials near the interface were increased. The maximum load, material hardness, and elastic modulus were positively correlated with the interface strength and interface thickness. The simulation provides a theoretical basis and research method for the study of grinding parameters and material removal mechanisms of C_f_/SiC composites.

Liao et al. [49] simulated the single grain scribing process, as shown in Figure 11. The chip formation and temperature field distribution of negative rake angle were simulated by FEM. The results show that the chip formation can be explained by cutting theory when the negative front angle is −15°~−40°. However, it is more appropriate to apply the hardness indentation principle proposed by Shaw [50] to describe the large negative rake angle. The shear angle decreases and the shear strain increases with the increase of the negative rake angle. Meanwhile, in large negative rake angle grinding, the highest temperature appears at the contact point between the abrasive tip and the workpiece, which is higher than the small negative rake angle grinding temperature. With the increase of the negative rake angle of abrasive particles, the energy consumed in chip formation increases. Wu et al. [51], through FEM, concluded that, under wet grinding conditions, grinding fluid can reduce the wheel sliding force (tangential force), which can reduce the grinding zone temperature. Long et al. [52] simulated and analyzed the temperature field in the process of high-speed grinding of engineering ceramics by using the FEM. The result showed that there was a high-temperature gradient in the shallow surface, which would generate a large thermal tensile stress and lead to machining defects such as thermal cracks.

Fang et al. [53] established a multi-interface stress transfer simulation of CMC materials by FEM, through a unit-cell model. The shear stress and optimized interface stress propagation were studied. As shown in Figure 12, at the middle position plane of the interface phase, the shear stress rapidly increases to the maximum value near the forced spot. Then, a decrease along the fiber direction occurs and tends to zero about 2R_f_ away from the forced spot. The above trends of shear stress are the same under different interfacial thicknesses, with the increasing thickness of the interface phase. The maximum value of shear stress gradually decreases. The position reaching the maximum value is gradually far away from the forced spot. In the radial direction of fiber, the shear stress exhibits a rapid increase near the interface between the fiber and the interface phase, reaching its maximum value at the interface. For the PyC interface phase, the shear stress decreases along the radial direction. After crossing the interfacial phase and the interface, the shear stress continues to decrease in the matrix along the radial direction. Liu et al. [54] established a scratching simulation model of SiC_f_/SiC based on the JH-2 model in ABAQUS (in Figure 13). The magnitude of the scratching force under different scratching speeds, depths, and different diamond types was analyzed, as well as the influence of fiber orientation. Li et al. [55] established a two-dimensional simulation model of a single diamond grain scribing C_f_/SiC in ABAQUS, and the influence of wheel speed and grinding depth on grinding force and workpiece surface morphology was analyzed. Ellahi et al. [56] analyzed the cutting performance of C_f_/SiC with a PCD tool by the FEM. In Figure 14, the fracture modes in parallel and vertical to the fiber direction and the corresponding force distribution near the cutting edge were analyzed. At cutting angle *θ* = 0°, with respect to the fiber direction, and the machining direction parallel to the fiber direction, the material is removed mainly due to the debonding of fiber and matrix. Initially, the matrix fracture and crushing of carbon fibers occur near the cutting edge, and cracks propagate along the fiber direction. In the uncut chip area, the bonding of fibers and matrix becomes weak as the fiber axial strength is far stronger than the bonding strength of fibers and matrix. Stresses mostly occur at the tooltip and propagate along the fiber direction. At a 90° fiber angle, the rake face of the tool is perpendicular to the fiber direction and carbon fibers are subjected to tensile stresses. Initially, cracks occur ahead of the tooltip and move in the feed direction. The shear strength of fibers is lower than axial tensile strength. Therefore, when fibers are subjected to the tensile stress, fibers begin to elongate and eventually break as the force surpasses the fibers’ shear strength. The friction between the cutting tool and workpiece is maximum at that point and the material has been removed mainly due to the shearing of fiber bundles. Stresses are mostly distributed at the rake face of the cutting tool and the maximum stress is concentrated around the tool edge. This research provides an efficient way to analyze the cutting performance of C_f_/SiC composite. 

However, all the above studies only established two-dimensional simulation models. CMC materials have a complex structure composed of matrix, fiber, and interface layer. The deformations are complicated during processing. The effect of weaving structure on material properties is very important. Huang et al. [57] established a 3D woven C_f_/SiC composite model and conducted a comparative study on single-grain scratching simulation with and without ultrasonic assistance. This research provides a reference for the establishment of geometric and constitutive models for carbon fiber, SiC ceramic matrix, and interface layer.

### 3.2. Smoothed Particle Hydrodynamics (SPH)

SPH is a meshless Lagrange method developed in 1977 [58]. It uses a group of particles to describe a continuous fluid (or solid). Each particle carries various physical quantities, including mass and velocity. By solving the dynamic equation of particles and tracking the trajectory of each particle, the mechanical behavior of the system can be obtained.

Unlike FEM, SPH can present precision and stable solutions to problems involving deformable boundaries, especially for large deformation and crack propagation, by approximating the governing equations of particles. SPH is more favorable for a cutting simulation owing to the particle-based algorithm, eliminating the element’s deformation between particles [59]. 

Liu et al. [60] built a three-dimensional single abrasive grain scratch model to analyze the SiC grinding mechanism, including the material removal process, speed effect, ground surface roughness, and scratching force by using SPH (in Figure 15a). The simulation results showed that the material removal process went through the pure ductile mode, brittle assisted ductile mode, and brittle mode with the increase of the cutting depth. The critical cutting depth of ductile–brittle transition was approximately 0.35 µm based on the variations in ground surface crack conditions, surface roughness, and maximum scratching force. And increasing the scratching speed promoted the transformation of deep and narrow longitudinal cracks into shallow and wide transverse cracks on the surface, which improved the surface quality. Duan et al. [61] constructed a three-dimensional FEM and SPH model of a single diamond scratching. As shown in Figure 15b, the 3D boundary was solved by coupling multiple SPH particles into an iron-solid cell consisting of multiple nodes. And in Figure 15c, Liang et al. [62] simulated the single grain scribing process with SiC ceramics by using SPH-coupled FEM. These studies provide effective methods with which to study grinding processes and ground surface quality for SiC and other brittle material.

Zheng et al. [63] developed an accurate and efficient modeling platform for simulating mechanical properties of particle-reinforced matrix composites. As shown in Figure 16, a master–slave method is adopted within SPH formulation for imposing the essential boundary conditions and other linear displacement constraints. It was proven that the optimized master–slave (MS) method can provide better accuracy for the displacement constraints. The optimization with modified boundary interpolation technique (i.e., MS2) is computationally as efficient as the PF method. This study provided a better choice for implementing essential boundary conditions and other displacement constraints to overcome the potential problems encountered by the penalty function (PF) method. Takabi et al. [64] instructed a method to build the SPH model of both ductile and brittle materials with damage criteria representing crack initiation and post-failure behavior for machining analysis. The effects of damage definition on the chip morphology and cutting forces demonstrate that an appropriate damage criterion must be taken into account for SPH cutting simulations, despite the natural separation of particles, and regardless of the ductility of the material. Although SPH is feasible in machining simulations, however, it underestimates forces, instabilities, and numerical issues with boundary particles, which can lead to inaccurate prediction. Shi et al. [65] simulated the grinding fracture process of the fiber and matrix of C_f_/SiC composites through the JH-2 model established by SPH. The microcracks initiation and propagation were analyzed and it was found that the radial crack was deflected due to the existence of carbon fibers. Zhou et al. [66] explored the grinding removal mechanism of 2.5D C_f_/SiC composites in different grinding directions based on experiments and SPH simulations, and found that the main removal mechanism is brittle fracture. The main damage modes are matrix cracking, interface debonding, fibers fracture, and fibers pulling out. And because of the existence of fibers, when the crack propagates towards the interface between fiber and matrix, the crack deflects. Grinding 2.5D C/SiC composites, the fibers fracture and fibers pulling out are obvious when the fibers’ direction is perpendicular to the feeding direction and the ground surface, and the ground surface roughness is the smallest in this direction. But in the other two directions, fiber fracture is the main factor, accompanied by a small amount of fibers pulling out. 

Despite its natural superiority in addressing the large deformation problems, SPH has low computational efficiency due to the kernel approximation compared to the conventional FEM.

### 3.3. Molecular Dynamics (MD)

In the 1990s, a large number of researchers began to study SiC ductile grinding. Due to the high experimental cost and long processing time in the ductile domain, many researchers began to consider the use of simulation methods, such as MD simulation, to explain the ductile removal mechanism of brittle materials.

MD analysis is a powerful tool for studying complex microscopic systems. This technique can obtain the motion trajectories of microscopic particles. MD analysis, as a theoretical research method, is very vital in studying nanoscale grinding process and has been successfully used to study the microscopic mechanism of tool wear, surface quality, and subsurface damage.

Based on the Tersoff [67] potential function, Li et al. [68] studied the effect of amorphous carbon interfacial layer thickness on the fracture mechanical behavior of SiC_NF_/SiC composites through MD calculations, and found that increasing the thickness of the interfacial layer would reduce the stress concentration coefficient of the fibers, increase the fracture energy, and transform the brittle fracture mode of the cracked penetrating fibers into the fiber pulling out failure mode, which can enhance the effect of reinforcing toughness, as shown in Figure 17.

Miao et al. [69] investigated the tensile mechanical properties of vertically aligned CNTs/SiC nanocomposites (VSNs) through an in-situ transmission electron microscopy (TEM) tensile test and MD simulation. As shown in Figure 18a, molecular models of VSNs used the Teroff empirical bond-order potential to describe interactions of SiC and interface between carbon nanotubes (CNTs) and the SiC matrix. And the interatomic forces in CNTs were described by the adaptive intermolecular reactive empirical bond order potential. During the tensile process, the periodic boundary condition was implemented in all directions and an NVT ensemble with a Nose–Hoover thermostat was employed to maintain a constant temperature of 300 K. Figure 18b,c shows the molecular models and results of tensile, pulling out, and peel-off of CNTs. The tensile mechanical performances of VSNs along the ‖ direction are higher than those along the ⊥ direction. Meanwhile, the fracture modes of VSNs are different along these two directions. The CNTs can bridge at the crack surfaces and further bear the stress transfer along ‖ direction. However, the weak interaction between CNTs and SiC matrix plays a key role in the tensile failure along the ⊥ direction. 

At present, the MD simulation is mainly applied to the cutting depth at the submicro and nano levels. And due to the maximum computing power being only a few cubic microns, the simulation scale is greatly limited. Meanwhile, the MD simulation is suitable for studying the movement of less than 10^−9^ s. So, the cutting speed used in the scratch experiment is only a few hundred microns per second, which cannot meet the speed requirements (several hundred meters per second) of the simulation experiment. In addition, in the MD simulation, the material must have a high deformation rate to ensure the accuracy of the calculation time, which is rarely used in the non-equilibrium state simulation of polymer processing models.

In summary, at the micro-scale, MD simulation methods are widely used in polymer structure design and composite material design. However, in the non-equilibrium state simulation for polymer processing, the calculation accuracy and force field model need to be improved.

### 3.4. Discrete Element Method (DEM)

During the machining process of the CMC, the interaction between wheel and workpiece gives rise to defects such as machining surface roughness and surface/sub-surface cracks. Experimental measurements or theoretical analysis of machining processes are difficult. The simulations like FEM and SPH are computationally expensive. Therefore, to deeply understand the grinding mechanism of the CMC, DEM is used to study the generation and propagation of micro-cracks during machining.

DEM was originally developed by Cundall and Stracke [70] in 1979 for the analysis of rock mechanics problems, and has been implemented in many other fields, such as DEM simulating particles of rocks, clayey soils, and ceramics.

Jiang et al. [71] simulated the grinding process of ceramics to understand the grinding mechanism, through building a two-dimensional particle flow program as a simulation platform. The process parameters used in the DEM model of the horizontal spindle plunge grinding process are shown in Figure 19a. The relationship among the grinding force and ground surface crack number with the grinding time is shown in Figure 19b, and the number of cracks and the average grinding force increase steadily with the increase of grinding time, and the grinding force of F_x_ increases faster than that of F_y_ due to the grinding chips filling the clearance of the grinding wheel. In addition, when the grinding force changes dramatically, the crack increases rapidly, resulting in the fracture of the ceramic parts. As shown by the red dot in Figure 19c, the grinding wheel moving forward, the crack growth, and the damage appear on the machining surface. Some micro-cracks can be found along the grinding track. The microcracks extend along the contact surface of the particles and form macroscopic cracks on the machined surface, resulting in the removal of the front-end material. It can be seen that the microcracks do not form obvious macroscopic intermediate cracks. The grains disintegrate into dispersed particles. In the grinding zone, most of the grains are crushed into particles, which are then quickly thrown away from the surface of the workpiece by the tangential force. Qiang et al. [72] established the DEM model of silicon carbide ceramics and the ultra-precision cutting model of the single point diamond. A dynamic simulation was carried out. The influence of the residual stress distribution on the direction of the workpiece depth under different cutting conditions, such as tool rake angle, cutting speed, and cutting depth, were analyzed. This study proved that a residual stress analysis is feasible with the DEM.

Li et al. [73], respectively, used DEM and the bonded particle model to establish and calibrate the discrete element models of the SiC ceramic matrix and carbon fiber. The displacement softening contact model was used to characterize the bilinear constitutive relationship of interlayer and fiber/matrix interface element damage. The production and expansion of the matrix crack and the dis-adhesion of the interface can be visually demonstrated. This study proved that the displacement-softened contact model can be used to study the elastoplastic of the interface of composite materials, which is feasible for studying C_f_/SiC composites with DEM.

DEM speed is fast and the storage space required is small, which is especially suitable for solving large displacement and nonlinear problems. In the DEM application of composite machining, the main issue lies in the fact that the mathematical formulation of this method requires a particular refinement. However, compared with the FEM, DEM is more advantageous in simulating the chipping of brittle matrix materials. On the contrary, stress and strain distribution estimations are normally less accurate.

### 3.5. Summary of Simulation Study

To sum up, all those numerical methods have been used to model machining processes. Typically, the FEM is used to simulate the micro-scale machining. The MD is used for the simulations of nanoscale machining processes. The length scale of the model dimensions in the DEM, which can be macroscale or micro-scale, depends on particle sizes and computational capacity. As discussed above, the DEM and MD simulations usually need to handle the interactions between massive numbers of particles and, hence, the computational cost can be extremely high when the total number of particles increases to a certain level. To maintain an efficient calculation, the total particle number should be appropriately selected. The model dimension should reasonably reflect the physical problem considered. A comparison between the FEM, SPH, DEM, and MD is shown in Table 2.

In addition, multiscale modeling combining atomistic simulation with continuum simulation to capture material deformation at different length scales has also been studied. The hybrid FEM-MD method is one of these methodologies. As shown in Figure 20, Wang et al. [74] proposed a multiscale method combining MD and FEM simulation to describe the separation behavior of the SiC/PyC interface and predict the stress–strain response in SiC_f_/SiC composites; it provides a way to constitute relation prediction.

## 4. Conclusions

In this paper, the research on CMC grinding technology is reviewed. Firstly, the research status of CG and non-CG technology is introduced, and their advantages and disadvantages are compared. Summarized in Table 1, those technologies have their own advantages, but there is still a lack of a process method that can simultaneously take into account the quality, efficiency, and cost of CMCs processing. Therefore, the theoretical research and application of the new multi-energy field composite machining method with high efficiency and high quality are still a research hotspot in the field of aerospace high-tech artificial manufacturing. Then, this paper briefly introduces four kinds of simulation methods from basic theory, research status, and application scope. The different characteristics, such as computing power and model scale, give the four simulation methods their own application scope and shortcomings. Similar to machining technology, it is a trend to apply the multi-scale simulation method to the study of CMCs machining. 

In addition to the studies on techniques and simulation methods, the particularity structure and physical properties of CMCs materials also deserve our attention The removal mechanism of CMC materials is complex and changeable due to their structural characteristics. Different physical properties, reinforcement, matrix materials, and the two-phase interface have diverse machinability, which change the removal mechanism. The matrix and reinforcing fibers are not necessarily removed synchronously during the grinding process. On the other hand, the anisotropy and filamentous toughening phase structure would cause random interface debonding and cracking events during a machining process. In addition, in the case of the particularity structure, the variation of fiber orientations relative to cutting directions, random waviness of individual fibers, and uneven fiber dispersion in the matrix bring about significant variations of material removal mechanisms during machining. Meanwhile, in terms of tool wear and machining rate, the high abrasion of the reinforcements to a cutting tool leads to excessive tool wear, which, in turn, brings about unsteady machining and significant subsurface damage such as delamination, reinforcement fracture, and burr formation, as well as varying surface integrity.

Therefore, the reinforced fibers not only improve the mechanical properties of the CMC materials, but also change the removal mechanism. Using new technology to improve processing quality, understanding the influence of fibers’ orientation and distribution on the material removal mechanism and removal mode, and establishing an appropriate mechanical processing model are important research directions for the future. The difficulties this research must face and the directions it may go in are given in Table 3. 

## Figures and Tables

**Figure 1 micromachines-15-00106-f001:**
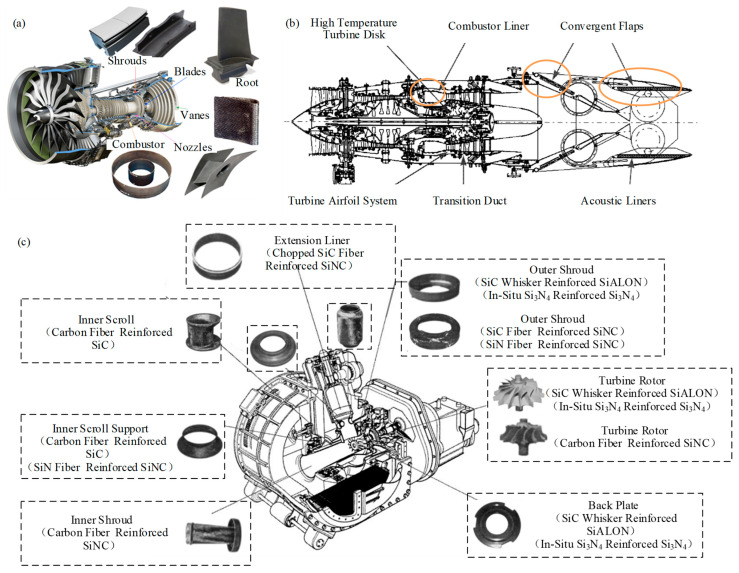
The application of CMCs: (**a**) high-temperature parts of aero-engines; (**b**) hot zone of the jet engine; (**c**) Main CMCs components of CGT.

**Figure 2 micromachines-15-00106-f002:**
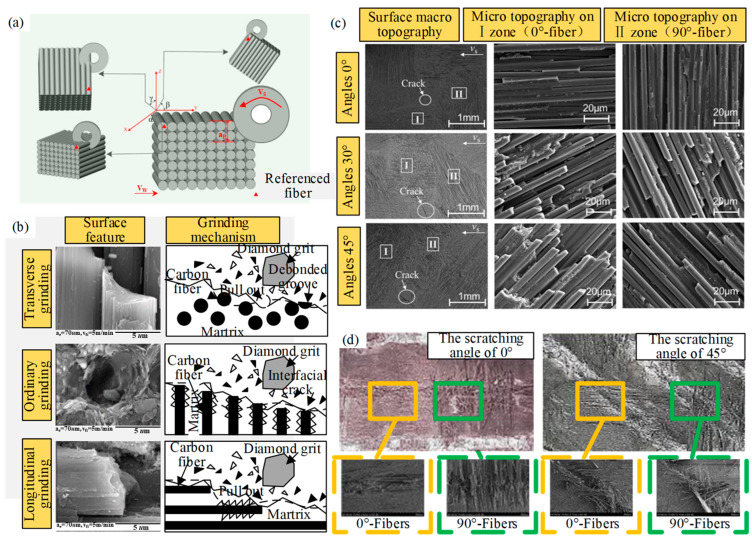
(**a**) Schematic diagram of the grinding direction and datum plane studied by Qu et al. [6]; (**b**) surface morphology and material removal mechanism of C_f_/SiC in transverse, normal, and longitudinal direction by Zhang et al. [5]; (**c**) SEM of the ground surface in different fiber orientations; (**d**) surface morphology of 0°and 90° fiber bundles at different scribing directions by Liu et al. [10].

**Figure 3 micromachines-15-00106-f003:**
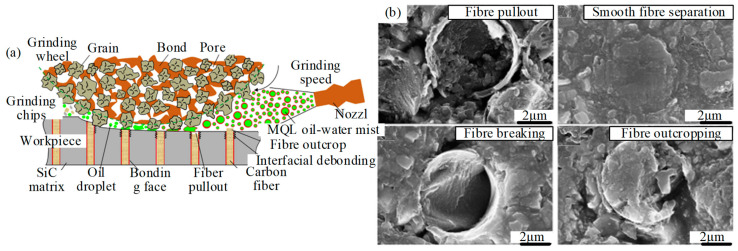
(**a**) Schematic diagram of the ground surface under the MQL system by Qu et al. [22]; (**b**) SEM photo of typical fracture of carbon fiber under MQL.

**Figure 4 micromachines-15-00106-f004:**
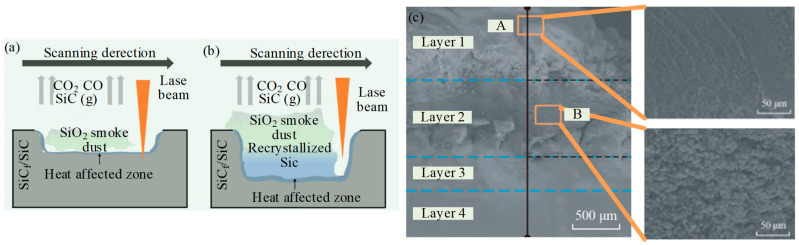
(**a**,**b**) Schematic diagram of laser-induced ablation under LAG; (**c**) the products of continuous laser ablating CMCs under LAG by An et al. [25].

**Figure 5 micromachines-15-00106-f005:**
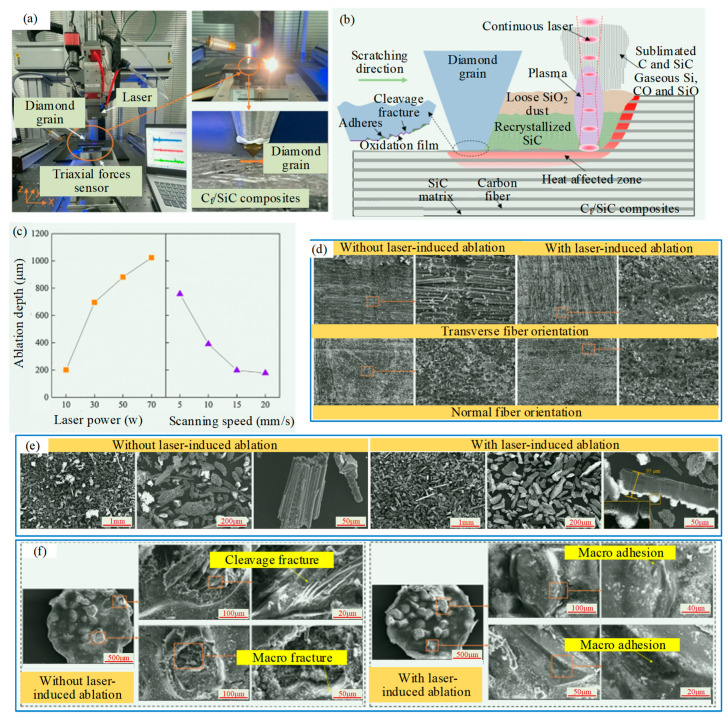
Laser-induced ablation-assisted grinding (LIAAG) by Zhou et al. [26]: (**a**) Laser-induced ablation-assisted grain scratching testing machine; (**b**) material removal mechanism of C_f_/SiC composites during LIAAG; (**c**) temperature field and ablation depth during laser ablation; (**d**) SEM images of C_f_/SiC composites surface after grinding: (**e**) SEM images of grinding chips; (**f**) wear morphology of diamond abrasive grains.

**Figure 6 micromachines-15-00106-f006:**
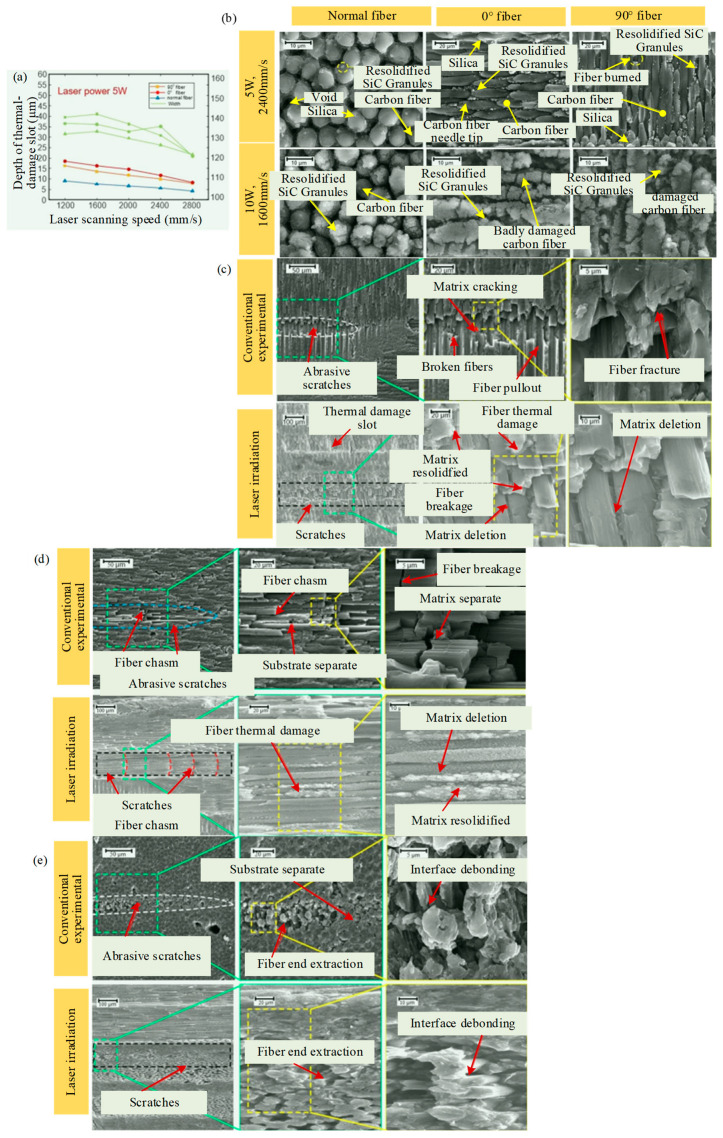
Laser-assisted grinding to improve the processing quality of 3D C_f_/SiC composites: (**a**) Relationship between laser power, scanning speed and thermal-damage slot size; (**b**) micro-structure of fiber surface after laser irradiation; (**c**) microscopic profile of 0° fiber area scratches; (**d**) microscopic profile of 90° fiber area scratches; (**e**) microscopic profile of normal fiber area scratches.

**Figure 7 micromachines-15-00106-f007:**
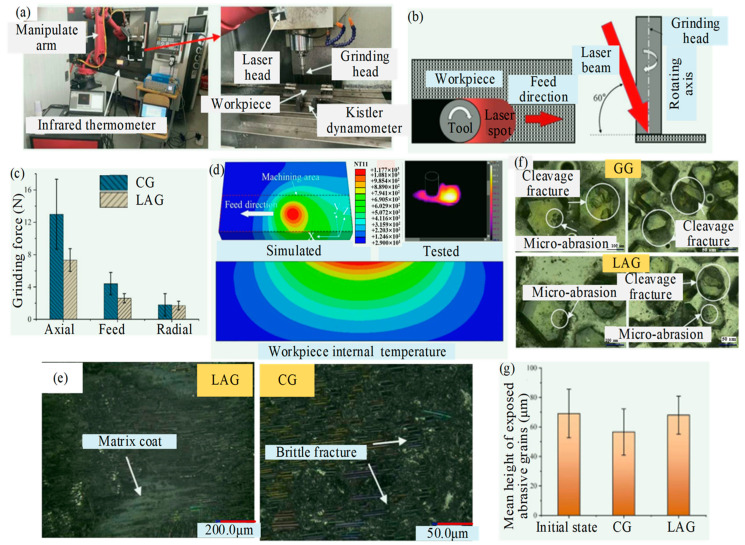
The cutting performance of CG and LAG of SiC_f_/SiC by Kong et al. [28]: (**a**) LAG system; (**b**) schematic diagram of LAG; (**c**) comparison of CG and LAG forces; (**d**) laser temperature field; (**e**) surface morphology; (**f**) grinding head abrasive grains wear forms; (**g**) mean heights of exposed abrasive grains.

**Figure 8 micromachines-15-00106-f008:**
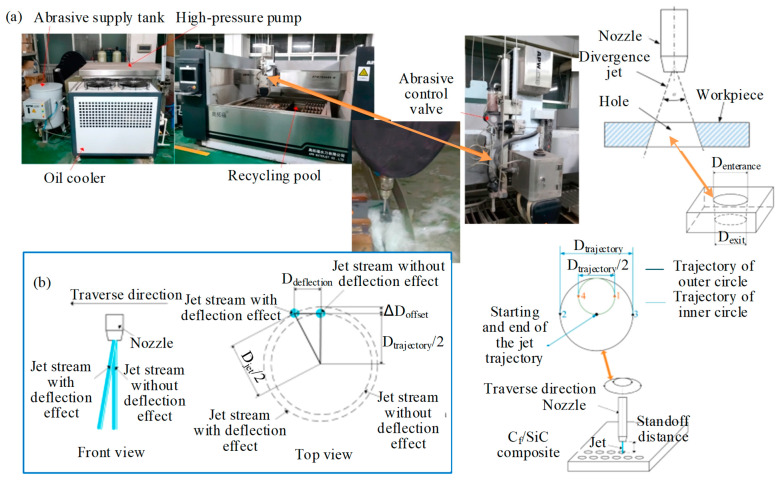
AWJM hole-making of C_f_/SiC by Zhang et al. [33]: (**a**) Experimental setup and processing diagram of AWJM; (**b**) schematic diagram of deflection effect.

**Figure 9 micromachines-15-00106-f009:**
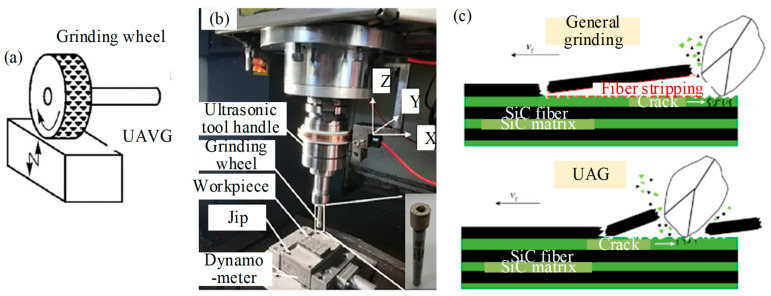
The experiment on UVAG of SiC_f_/SiC material by Kang et al. [37]: (**a**) a type of UVAG; (**b**) UVAG test platform; (**c**) schematic diagram of grinding removal process of SiC_f_/SiC composites.

**Figure 10 micromachines-15-00106-f010:**
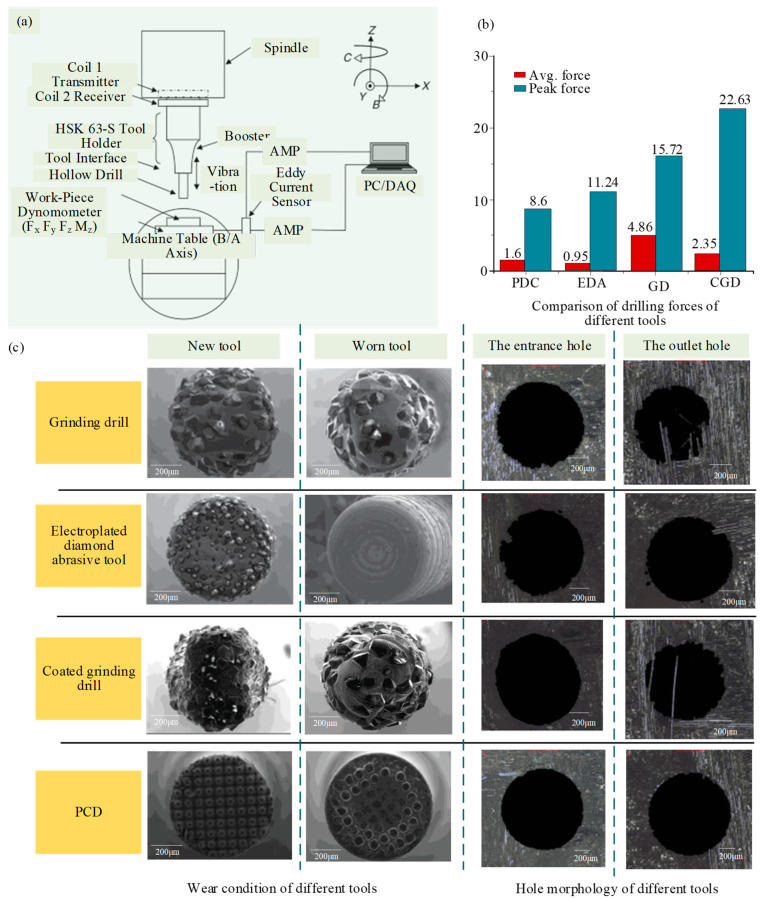
Different tools for micro-hole drilling of SiCf/SiC CMC assisted by ultrasonic vibration by Bertsche et al. [40] and Huang et al. [43]: (**a**) Experiment setup; (**b**) Comparison of drilling forces of different tools; (**c**) Hole morphology and wear condition of Different tools.

**Figure 11 micromachines-15-00106-f011:**
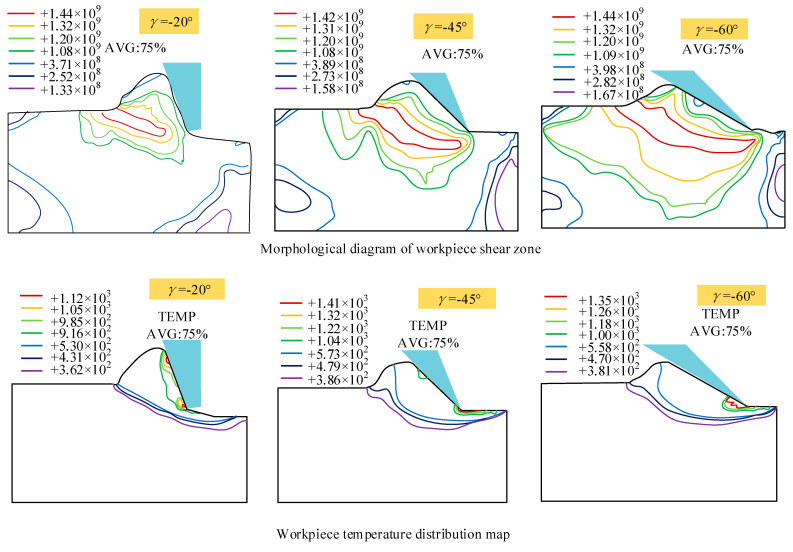
The morphology and temperature distribution of the workpiece shearing zone during abrasive grinding.

**Figure 12 micromachines-15-00106-f012:**
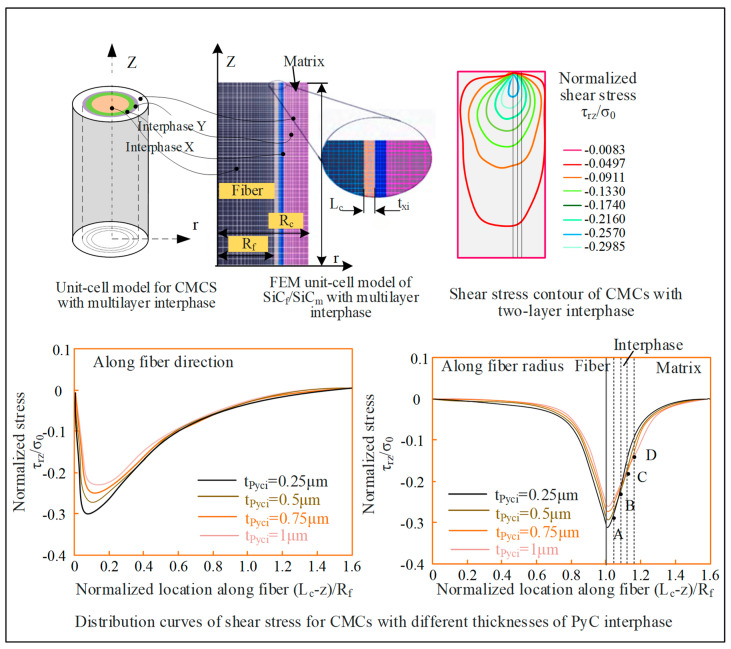
Multi-interface stress transfer simulation of CMC materials.

**Figure 13 micromachines-15-00106-f013:**
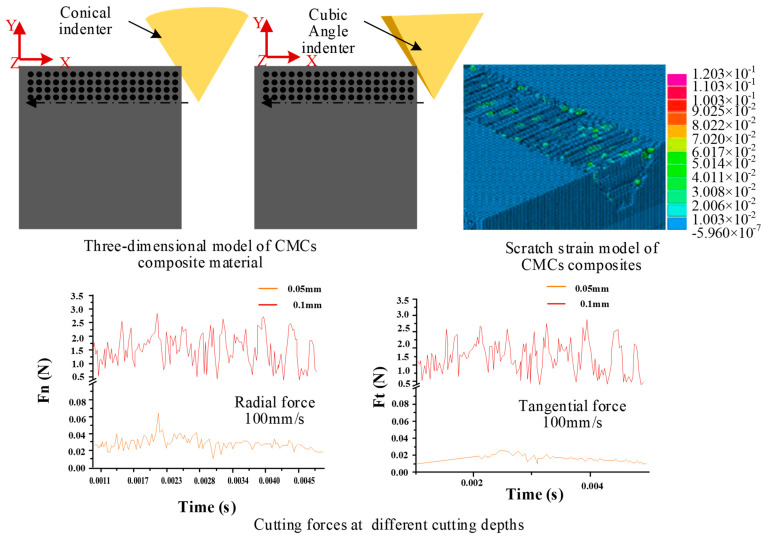
Scratching simulation of SiC_f_/SiC based on JH-2 model.

**Figure 14 micromachines-15-00106-f014:**
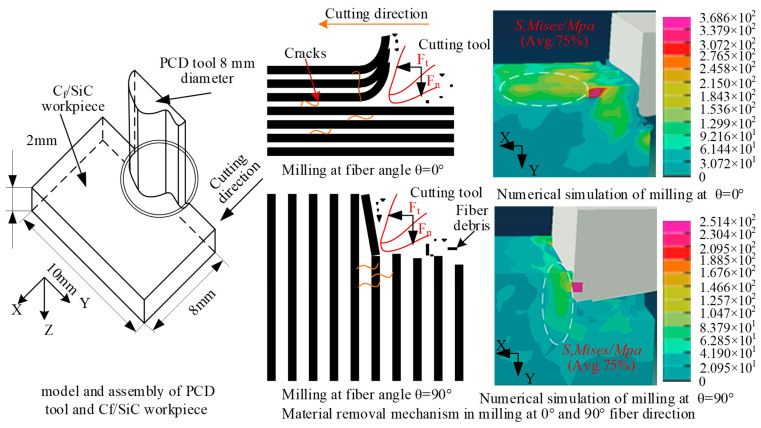
C_f_/SiC cutting simulation with PCD tool.

**Figure 15 micromachines-15-00106-f015:**
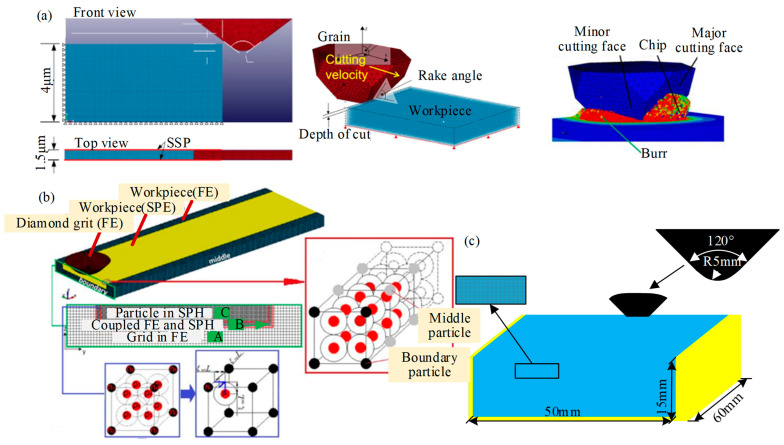
SPH model of single abrasive grain grinding silicon carbide ceramics. (**a**) Schematic for scratching model by Liu et al. [60]; (**b**) The 3D simulation model for single diamond scratching by Duan et al. [61]; (**c**) SPH grinding model before scratching by Liang et al. [62].

**Figure 16 micromachines-15-00106-f016:**
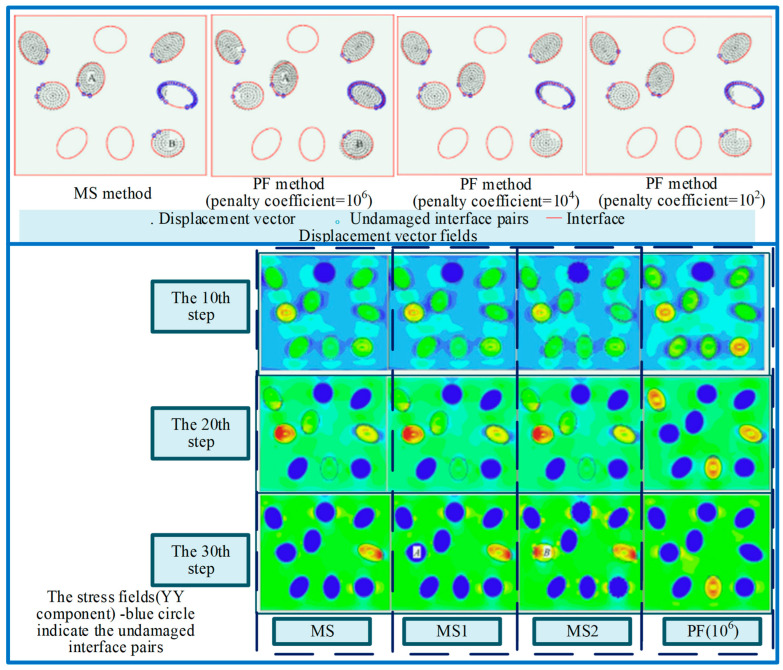
Modelling for particle-reinforced matrix by Zheng et al. [57].

**Figure 17 micromachines-15-00106-f017:**
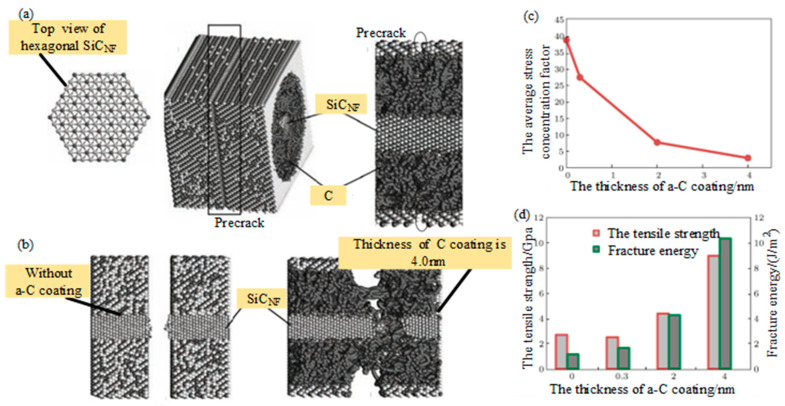
The effect of amorphous carbon interfacial layer thickness on the fracture mechanical behavior of SiC_NF_/SiC composites through MD calculations studied by Li et al. [68]: (**a**) Atomic configuration of SiC_NF_/SiC nanocomposite with predefect in matrix; (**b**) atomic configuration of fracture surface in SiC_NF_/SiC nanocomposite with C coating in different thickness; (**c**) the average stress concentration factor of SiC_NF_ vs. the thickness of C coating; (**d**) the tensile strength and fracture energy of SiC_NF_/SiC nanocomposite as a function of the thickness of C coating.

**Figure 18 micromachines-15-00106-f018:**
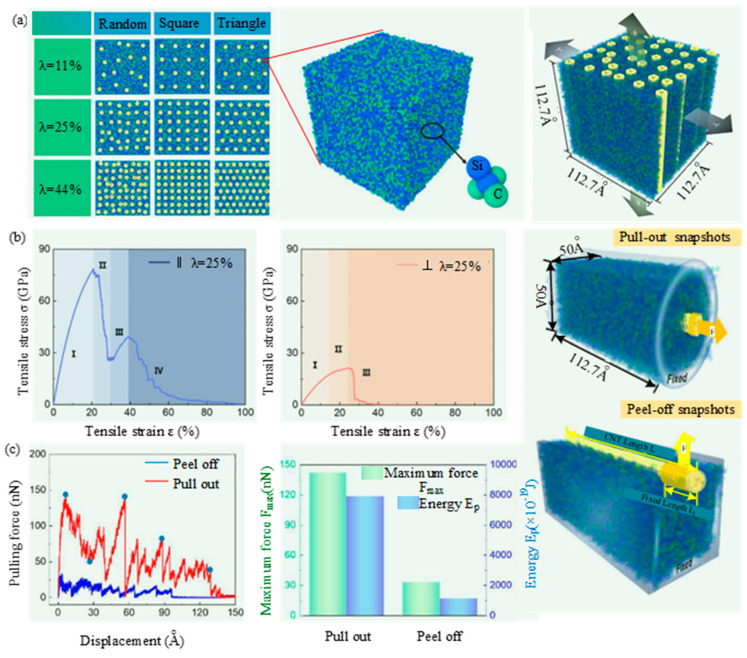
The tensile mechanical properties of vertically aligned CNTs/SiC nanocomposites (VSNs) through MD simulation by Miao et al. [69]: (**a**) Molecular models of VSNs with different carbon nanotubes (CNTs) distribution types and contents and tensile models of VSNs along ‖ and ⊥ directions; (**b**) tensile stress–strain curves and tensile snapshots with stress distribution along ‖ and ⊥ directions; (**c**) molecular models for pull-out and peel-off of CNTs from SiC matrix.

**Figure 19 micromachines-15-00106-f019:**
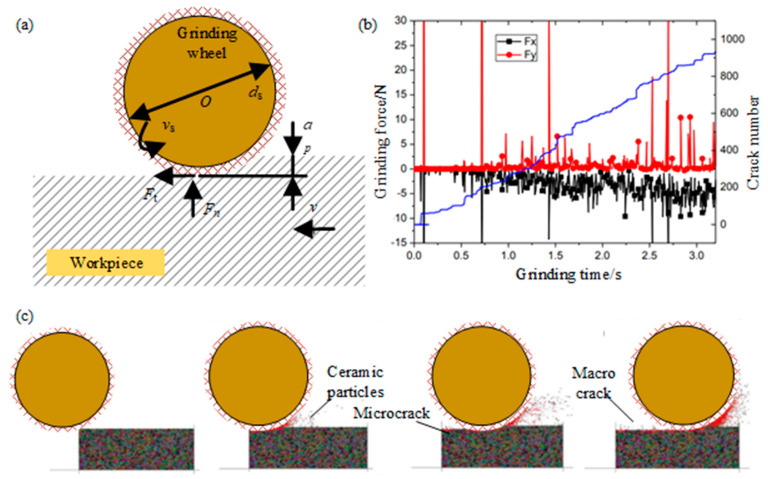
(**a**) DEM model for SiC ceramics; (**b**) simulation of grinding process; (**c**) relationship between grinding force and crack number by Jiang et al. [71].

**Figure 20 micromachines-15-00106-f020:**
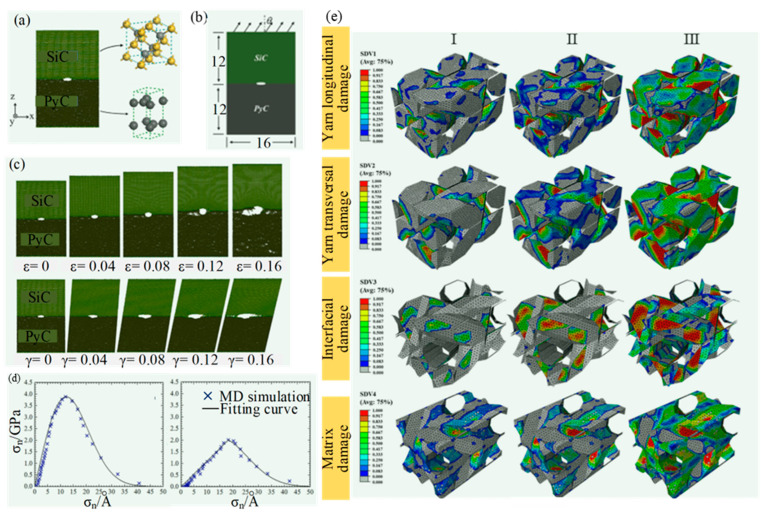
The hybrid of FEM-MD method by Wang et al. [74]: (**a**) MD simulation model of SiC/PyC interface; (**b**) load on MD simulation model; (**c**) damage evolution process of interface crack under pure tensile and shear load; (**d**) on the basis of MD simulation, the traction-separation laws (TSLs) for normal and tangential in interface; (**e**) damage evolution of 3D woven composite FEM model (Ⅰ: ε_Z_ = 0.03%; Ⅱ: ε_Z_ = 0.23%; Ⅲ: ε_Z_ = 0.75%).

**Table 1 micromachines-15-00106-t001:** Comparison of different CMC grinding technologies.

Method	CG	LAG	UVAG	AWJM
Advantage	Wide application range, simple process, and high processing efficiency.	Reduce material hardness and improve processing performance.	Good processing quality, small surface loss.	Super-hard abrasive high-speed impact workpiece surface to achieve removal processing, no thermal effect.
Shortcoming	Rely on diamond tools, which are expensive and prone to serious wear; and It is difficult to process parts with complex shapes and high dimensional accuracy.	The heat-affected zone material must be removed to obtain the desired surface; High temperature will reduce the cutting performance of the tool, in particular lead to diamond graphitization.	The processing efficiency is low and the processing range is limited.	Large impact force, easy to break the edge and damage the surface of the workpiece.

**Table 2 micromachines-15-00106-t002:** A comparison between the FEM, SPH, DEM, and MD.

Method	FEM	SPH	DEM	MD
Basic theory	Continuum mechanics	Meshless Lagrange method, Kernel Function and Describing a continuous fluid (or solid) with a swarm of interacting particles	Newton’s law of motion and the relationship between force and relative displacement between neighbor particles	Newton’s law of motion and the potential function
Timestep	~μs	~μs	~μs	~fs
Length scale	Macroscale to mesoscale	Macroscale to mesoscale	Macroscale to mesoscale	Nanoscale
Common usage	Continuum materials and composites	Continuum materials and composites	Granular and discontinuous materials, composites	Nanomaterials
Limitation	Cannot well represent the discreteness, fracture, and damage processes in materials	It requires a relative high computation and a lot of time to calibrate the parameter	Requires a relatively high computation and a lot of time to calibrate the parameter	Requires a huge amount of computation and limits to nanometric sizes

**Table 3 micromachines-15-00106-t003:** Research difficulties and direction for grinding technology for SiC CMC materials.

Main Difficulty	General Problem	Research Directions
Diverse machinability	The great difference in physical properties between SiC matrix and C fiber, and the matrix and reinforcing fiber are not necessarily removed synchronously.	The effect of reinforcing fibers on the removal mechanism and mode of the material.
The anisotropy and filamentous toughening phase structure	The reinforcement-matrix interphase material with random interface debonding and cracking events during a machining process.	The mechanical model of the two-phase interface and the transformation of the removal mode at the bonded surfaces during grinding.
Fiber orientation	The variation of fiber orientations relative to cutting directions, random waviness of individual fibers, and uneven fiber dispersion in the matrix bring about significant variations of material removal mechanisms.	Grinding mechanism and surface quality on different fiber orientations.
Tool wear	The high abrasion of the reinforcements to a cutting tool leads to excessive tool wear.	Application of non-CG and research on new grinding mechanism; research on grinding quality under different grinding tools.

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
