# Peer review of "Review on Abrasive Machining Technology of SiC Ceramic Composites"

_micromachines, 2024, doi:10.3390/mi15010106_

Round 1

Reviewer 1 Report

Comments and Suggestions for Authors

In the manuscript titled “ Review on Abrasive Machining Technology of SiC Ceramic Composites”, the recent abrasive machining experimental study of CMC abrasive machining technologies, including the conventional grinding (CG), laser-assisted grinding (LAG), abrasive waterjet machining (AWJM), and ultrasonic vibration-assisted grinding (UVAG), were reviewed to reveal the material removal mechanism. Moreover the commonly used numerical simulation research for modeling and simulating the machining of ceramic matrix composites were briefly summarized. This manuscript introduces the current trend and pivotal points of abrasive machining of CMC, which has good guidance and reference significance for readers. This review is accessible to the average researcher, and can arouse the interest and attention of the readers concerned. However, there are some minor problems, which must be solved before it is considered for publication. If the following problems are well-addressed, this manuscript is recommended for employment.

1. Although the author has made a summary of Ultrasonic vibration-assisted grinding in the section 2.4, the studies you reviewed are not comprehensive. This part needs more detailed research and summary .

2. In page 18, line 3 of section 3.4 , “FEA” is short for what? I guess the author means “FEM” rather than “FEA”. The same situation appears in the title of Table 2.

3. In page 20, line 10 of section 3.5, the hybrid FE-MD method maybe FEM-MD method.

Comments on the Quality of English Language

Minor editing of English language required

Reviewer 2 Report

Comments and Suggestions for Authors

This literature review paper provides a meticulous and comprehensive examination of abrasive machining techniques applied to ceramic matrix composites. The author has demonstrated a thorough understanding of the subject, offering a detailed and insightful analysis of relevant literature in this field.

The current reviewer raises valid concerns regarding certain judgmental statements found within the manuscript. It is advisable to critically evaluate and potentially revise these statements to ensure that the language used maintains an objective and unbiased tone. The manuscript's overall credibility and neutrality would benefit from addressing these concerns

For instance, the statement highlighting the drawbacks of the Ultrasonic Vibration-Assisted Grinding (UVAG) process, specifically mentioning 'low materials removal rate' and 'complex and expensive machining costs of ultrasonic generators,' lacks specific references or citations. It is essential to ensure that such judgmental statements are supported by concrete evidence from the literature. Moreover, it's important to note that the main limitation of UVAG (especially rotary spindle) is typically associated with its suitability for hole making, and not its overall cost-effectiveness or complexity. The manuscript should provide a more nuanced and accurate representation of the advantages and limitations of UVAG, acknowledging its cost-effectiveness in certain applications while highlighting its constraints, such as its limited adaptability for cutting different shapes and dimensions beyond round holes.

It is recommended that the authors carefully review and validate all of alike judgmental statements, ensuring that they are supported by relevant citations and references. This will not only enhance the overall credibility of the manuscript but also provide readers with a clear and substantiated understanding of the discussed points. Additionally, the authors are encouraged to critically assess the accuracy of each statement, considering potential nuances or alternative perspectives within the literature
